# Learning from a Sample in Online Algorithms

**C.J. Argue**     **Alan M. Frieze**     **Anupam Gupta**     **Christopher Seiler**
Departments of Computer Science and Mathematics
Carnegie Mellon University
Pittsburgh, PA 15213

## Abstract

We consider three central problems in optimization: the restricted assignment load-balancing problem, the Steiner tree network design problem, and facility location clustering. We consider the online setting, where the input arrives over time, and irrevocable decisions must be made without knowledge of the future. For all these problems, any online algorithm must incur a cost that is approximately $\log |I|$ times the optimal cost in the worst-case, where $|I|$ is the length of the input. But can we go beyond the worst-case? In this work we give algorithms that perform substantially better when a $p$-fraction of the input is given as a sample: the algorithm use this sample to *learn* a good strategy to use for the rest of the input.

## 1   Introduction

The area of sequential decision-making (a.k.a. online algorithms) is a central direction in algorithms under uncertainty: faced with a sequence of requests arriving over time, an algorithm needs to make decisions without knowing the future. Given this lack of information, how can it make near-optimal decisions? An enormous amount of work has been done on problems in this area over the past decades, and we know near-optimal worst-case bounds for many problems of interest in clustering, online resource allocation, scheduling, and network design.

However, these worst-case bounds are often pessimistic (perhaps by definition): they focus on the hardest, pathological instances. Can we give algorithms that work for instances that arise in practice, which tend to be very far from the worst case? There have been many attempts to formalize this question: see the textbook by Roughgarden [2020] for several approaches to go beyond the worst-case.

In this paper we focus on *online algorithms with a sample (AOS)* framework of Kaplan et al. [2020, 2022]: a similar model was defined by Lattanzi et al. [2021] based on Kumar et al. [2019]. In this model, we assume that an arbitrary instance $I$ is chosen, perhaps by an adversary. A random sample $S$ containing a $p$-fraction of the instance is then presented to the algorithm, as part of a "training phase". Having used this sample to learn something about $I$, the algorithm faces the rest of the instance $R := I \setminus S$ (in its adversary-chosen order). This is essentially the "testing phase". The algorithm needs to service these requests, and ensure that for each input $I$,

$$\mathbb{E}_R[\text{ cost of algorithm } \mathcal{A} \text{ on } R] \leq g(p) \cdot \mathbb{E}_R[\text{ optimal cost for } R]. \tag{1}$$

Here the expectation is taken over both the $p$-random sample $S$ (and hence its complement $R$), as well over the internal randomness of the algorithm $\mathcal{A}$. Can we design algorithms that use this small random sample (just an $p$ fraction) to extract enough information to substantially improve its performance?

We show the answer is "yes" for three combinatorial optimization problems in load-balancing, clustering, and network design:

36th Conference on Neural Information Processing Systems (NeurIPS 2022).

1. **Load-Balancing**: Jobs arrive over time; each job $j$ can be assigned to one of some subset $N(j) \subseteq [m]$ of some $m$ fixed machines. Minimize the maximum load of any machine.

2. **Clustering**: In *facility location*, clients arrive in a metric space, and we want to cluster them (and open one facility in each cluster). Minimize the total cost to open facilities and connect clients to them.

3. **Network Design**: In *Steiner tree*, points of a metric space arrive one by one, and need to be connected up to form a spanning tree. Minimize the total length of the tree edges.

All three problems have been studied in the worst-case setting, where it is known that for any online algorithm there are instances where the algorithm's cost must be least $\approx \Omega(\log |I|)$ factor larger than the optimal cost; here $|I|$ is the size of the instance. In this paper we want guarantees which do not depend on the instance size $|I|$, but only on the fraction $p$ of the input in the sample.

**Our Contributions.** Our main results are that given an $p$-sample, the performance improves considerably. For the load-balancing problem, we show that our LEARN-&-LOAD algorithm can achieve a load of

$$(1 + \varepsilon)\, \mathbb{E}[\mathrm{Opt}(R)] + O\left(\frac{\log^3 m}{p\varepsilon^4}\right) \tag{2}$$

for any $\varepsilon > 0$, where $m$ is the number of machines. Hence, if the optimal load is not too small (at least poly-logarithmic in the number of machines) then an online algorithm can achieve a load that is arbitrarily close to this optimal load. Contrast this with the worst-case setting (i.e., without a sample), where such a result is provably impossible: for any online algorithm we can give instances $I$ where $\mathrm{Opt}(I)$ is as large as we want, and yet the algorithm's load is at least $(\log_2 m) \cdot \mathrm{Opt}(I)$. So having access to the random samples is provably very powerful.

For clustering (facility location) and network design (Steiner tree), we get algorithms whose cost is

$$O(\log 1/p) \cdot \mathbb{E}[\mathrm{Opt}(R)].$$

So when $p > 0$ is a constant, say, our cost no longer depends on the size of the input. Indeed, setting $p := 1/|I|$ means we get a sample of a single element (which is conceptually like not having any sample at all), and we match the worst-case bounds. In this sense, our results interpolate between the AOS framework and the worst-case.

**Our Techniques.** One of the main contributions of the LEARN-&-LOAD algorithm is conceptual: we show that a *low-round parallel algorithm* to compute a suitable parameter vector $\theta$ can be used to get a generalization bound. Let us explain. We use an observation of Agrawal et al. [2018], Lattanzi et al. [2020] that for any load-balancing instance $R$, there exists a weight vector $\theta \in \mathbb{R}^m$ such that assigning each job $j$ fractionally to machines $i \in N(j)$ using the proportional allocation $x_{ij} := \frac{\theta_i}{\sum_{i' \in N(j)} \theta(i')}$ achieves an essentially optimal allocation. Now we could try to use PAC-style arguments: compute the empirical risk minimizer using the sample $S$, and use it for the real requests $R$. Then apply uniform-convergence bounds to show generalization. Applying this idea gives a bound of

$$(1 + \varepsilon)\mathbb{E}[\mathrm{Opt}(R)] + \frac{m\,\mathrm{poly}(\log m/\varepsilon)}{p}.$$

Notice that this result suggests we need the optimal load to be very large—larger than the number of machines $m$—for it to exceed the second term and thereby get non-trivial results. This is undesirable: we want to see improved results for much smaller values of $\mathrm{Opt}$ (like the bound of (2) we will eventually show), and PAC-style bounds on covering numbers don't seem to give such tighter bounds. (Bounds on covering numbers seem to need an exponential dependence on the dimension $m$, which leads to $\mathrm{poly}(m)$ bounds.)

A different approach would be to view the maximum load of an instance $R$ as a function of $\theta$, and use the requests from the sample phase $S$ to implement SGD update steps, whence the convergence guarantees of stochastic gradient descent by Nemirovski and Yudin [1983] would imply generalization bounds [Hardt et al., 2016, Hardt and Recht, 2021]. However, this approach fails because the maximum load of $R$ (as a function of $\theta$) is neither concave nor convex.

Our approach is different: we show that (a) a multiplicative-weights update algorithm to compute an optimal $\theta$ for the instance $R$ requires only a *poly-logarithmic* number of rounds (based on the work

of Ahmadian et al. [2021]), and (b) each round can be implemented using a *robust load query* to the instance $R$. The eventual idea becomes very clean: we partition the sample $S$ into poly-logarithmic mini-samples, and use each mini-sample to implement one robust load query to $R$. (Here we use that $S$ and $R$ are both samples from $I$.) We hope this idea of using a low-round algorithm to compute $\theta$ as a learning algorithm might prove useful in other contexts.

The algorithm for clustering and network design follow a different conceptual message: that of using (a suitably scaled version of) the sample $S$ to build an anticipatory solution $F$, and to then augment this solution $F$ to get the solution for the real requests $R$.

## 1.1 Related Work

The AOS model was formalized by Kaplan et al. [2020, 2022]; a closely related model was given by Correa et al. [2021], who consider the items to be independently sampled with probability $p$ (instead of a random $pn$-sized subset being chosen as the sample). Independently, Lattanzi et al. [2021] considered the *semi-online* setting of Kumar et al. [2019] where part of the input is given offline and the rest is online; they studied the case where a *random $p$-fraction* is presented offline. They consider the correlation clustering problem and show how the AOS model overcomes strong impossibility results. One difference between these two models is that the semi-online model seeks to bound the algorithm's cost on $\mathcal{I}$ to $\mathrm{Opt}(\mathcal{I})$, i.e., instead of (1), it aims to show that

$$\mathbb{E}_R[\text{ cost of } \mathcal{A} \text{ on } \mathcal{I}] \leq h(p) \cdot \text{ optimal cost for } \mathcal{I} . \tag{3}$$

We focus on definition (1), but our techniques are malleable and extend to definition (3) as well.

Similar models have been studied in other contexts. In the *random-order model* and the *secretary* problem from optimal stopping theory widely used in online algorithms and algorithmic mechanism design, the input is adversarially chosen but presented to the algorithm in an uniformly random order. The major difference from the AOS model is that the second part $R$ also appears in random order. However, some algorithms (which are called *order-oblivious* by Azar et al. [2014]) for the random-order model do not use this randomness of $R$'s order, and hence can be implemented in the AOS model as well. For example, the algorithm of Dynkin [1963] for the secretary problem, and the algorithms of Devanur and Hayes [2009], Agrawal et al. [2014], Molinaro and Ravi [2012] for online allocation problems; see Gupta and Singla [2020] for a discussion of random-order models.

Our work falls in the broad area of ML-Augmented Algorithms (see, e.g., the survey by Mitzenmacher and Vassilvitskii [2020] or the works by Mahdian et al. [2007], Lykouris and Vassilvtiskii [2018]), where the algorithm takes some (machine-learned) advice and uses this to give an improved performance. Here we crucially address the question of *how to learn this advice* from the given data. This has been done in a handful of previous works, e.g., by Devanur and Hayes [2009], Agrawal et al. [2014], Molinaro and Ravi [2012], Lavastida et al. [2021] for order-oblivious algorithms for online allocation, but these works use PAC-style ideas which we try to go beyond. Indeed, we hope our work further focuses attention on the learnability of the ML advice in these settings.

The idea of proportional allocation for load-balancing and matchings was first proposed by Agrawal et al. [2018], and extended by Lattanzi et al. [2020], Li and Xian [2021]. Ahmadian et al. [2021] gave a distributed algorithm to compute a weight vector $\theta$ which gives small maximum load with a small number of rounds of communication; we present an arguably simpler version of their arguments in Section 2.5, and show it also works when we are just given noisy statistical access to the instance.

The online Steiner tree problem was proposed by Imase and Waxman [1991], who gave optimal algorithms and matching lower bound results. The online facility location problem was proposed by Meyerson [2001], and the upper and lower bounds were improved by Fotakis [2008]. The algorithm in Meyerson [2001] also works in the random order model, but it is not order-oblivious. Garg et al. [2008] show that Steiner tree does not have any improved algorithms in the random-order model; in contrast they give constant-competitive algorithms when the input consists of i.i.d. samples from a known distribution; Dehghani et al. [2018] claim an extension to non-identical distributions. Our algorithm for Steiner tree in the sampling model is inspired by theirs ("build a solution on the sample, and then augment"). Azar et al. [2022] give a framework for network design with predictions; their ideas imply an alternative $O(\log 1/p)$-competitive guarantee for Steiner tree (using a more involved algorithm than ours). Their approach does not seem to extend to facility location or load balancing.

# 2 Load-Balancing with a Sample

In the online *load-balancing* problem (sometimes called the *restricted assignment model with parallel machines*) the input consists of $m$ machines which are fixed, and the online request contains $n$ jobs arriving over time. Each job specifies a subset of the $m$ machines that it be assigned to. (We consider unit-sized jobs and identical machines for simplicity, see §2.4 for extensions.) The algorithm must assign the job to one of these machines. The load of a machine is the total size of jobs assigned to it, and the goal is to minimize the maximum load over all machines.

It is convenient to think of machines being the vertices of a hypergraph $H$, and denoting them by $u, v$, etc.; let $V$ denote the set of machines, with $|V| = m$. Each job is a hyperedge $e \subseteq V$, containing the machines it can be assigned to. Let $E$ denote the set of jobs, with $|E| = n$. We seek an assignment $\varphi : E \to V$ satisfying $\varphi(e) \in e$ for each job/edge $e \in E$; the load of machine $v \in V$ is $|\varphi^{-1}(v)|$. The goal is to minimize $\mathbb{E}[\max_v |\varphi^{-1}(v)|]$, the (expected) maximum load over all machines, where the expectation is taken over any random choices we make in finding $\varphi$.

Given an instance $\mathcal{I}$ of load-balancing, a *fractional assignment* of these jobs is a matrix $\mathbf{X} \in \mathbb{R}^{n \times m}$ given by the following linear program:

$$P_{LB} := \{ \quad \begin{array}{ll} \sum_{v \in E} x_{ev} = 1 & \forall e \in E \\ \sum_{e : v \in e} x_{ev} \leq L & \forall v \in V \\ x_{ev} \geq 0 & \}. \end{array}$$

The smallest value of $L$ for which this LP is feasible is the *optimal fractional load* $\mathrm{FOpt}(\mathcal{I})$; if we restrict $x_{ev}$ to integers, then this smallest value of $L$ is the *optimal (integer) load* $\mathrm{Opt}(\mathcal{I})$. Both can be found *offline* using flow techniques, though our focus is in the online setting. Henceforth we use the notation $\mathsf{load}(\mathbf{X}) := \|\sum_e x_e\|_\infty$, where $x_e \in \mathbb{R}^m$ is the row of $\mathbf{X}$ corresponding to the job $e$, denoting its fractional assignment.

Our approach is to compute a fractional assignment $\mathbf{X}$ online, and use independent randomized rounding to convert it (also online) into an integer assignment $\varphi$.

**Lemma 2.1.** *Given a fractional $\mathbf{X}$, randomized rounding outputs an (integer) assignment $\varphi$ with*

$$\mathbb{E}\big[ \max_v |\varphi^{-1}(v)| \big] \leq (1 + \varepsilon) \, \mathsf{load}(\mathbf{X}) + O(\varepsilon^{-2} \log(m/\varepsilon)).$$

*Proof.* Randomized rounding sends job $e$ to a machine $v$ independently with probability $x_{ev}$: the expected load of each machine $v$ is $\mu_v := \sum_e x_{ev}$, and a concentration bound implies that

$$\Pr\big[ |\varphi^{-1}(v)| \geq (1 + \varepsilon/2) \, \mathsf{load}(\mathbf{X}) + O(\varepsilon^{-2} \log(m/\varepsilon)) \big] \leq \frac{\varepsilon}{2m^2}.$$

Taking a union bound over all $m$ machines increases the failure probability to at most $\frac{\varepsilon}{2m}$. Finally, the maximum load is at most $n$ and the fractional load must be at least $n/m$, so the contribution to the expectation in the event of failure is at most is at most $\frac{\varepsilon}{2m} \cdot n \leq \varepsilon/2 \, \mathsf{load}(\mathbf{X})$. $\square$

Given this rounding algorithm, we can focus on finding a fractional assignment online with low $\mathsf{load}(\mathbf{X})$ close to $\mathrm{FOpt}(\mathcal{I}) \leq \mathrm{Opt}(\mathcal{I})$. The randomized algorithm above can be derandomized, and also better additive losses can be obtained, see work by Li and Xian [2021]; however, these change only the lower order terms in our proofs.

## 2.1 The Proportional Allocation Strategy

To find a fractional assignment, we use an important idea by Agrawal et al. [2018], Lattanzi et al. [2020]. They show that for any instance $\mathcal{I}$ and any threshold $\lambda > \mathrm{FOpt}(\mathcal{I})$, there exists a *weight vector* $\theta \in \mathbb{R}_+$ such that the fractional assignment

$$x_{ev} := \frac{\theta_v}{\sum_{u \in e} \theta_u}$$

achieves $\mathsf{load}(\mathbf{X}) \leq \lambda$. We denote the vector of individual machine loads corresponding to this $\theta$, and the maximum load, as

$$L(\mathcal{I}, \theta) := \mathbf{1}^\intercal \mathbf{X} = \sum_e x_e, \quad \text{and} \quad \mathsf{load}(\mathbf{X}) = \|L(\mathcal{I}, \theta)\|_\infty.$$

While $\theta$ can be found using a convex program, this does not help us prove generalization bounds for the AOS model. In the rest of this section we show that (a) the natural multiplicative-weights algorithm to compute $\theta$ converges in $O(\log^2 m/\varepsilon^2)$ rounds, even with restricted and slightly noisy access to the instance $\mathcal{I}$, and then (b) how this result gives an algorithm in the AOS model.

To model the restricted/noisy access to the input $\mathcal{I}$, we consider the following framework:

**Definition 2.2** (Robust Oracle). *Given any function $f$ and a parameter $k$, a $(k, \varepsilon)$-robust oracle for an instance $\mathcal{I}$ returns a value $V$ such that*

$$(1 + \varepsilon)^{-1} \cdot \max(f(\mathcal{I}), k) \leq V \leq (1 + \varepsilon) \cdot \max(f(\mathcal{I}), k).$$

In fact, we will consider such an oracle only for statistical queries, where we apply a function $f$ to each element of the dataset $\mathcal{I}$ (in our case the jobs/edges of the instance), and then return their sum, so this can be viewed as a variant on the well-known SQ model of Kearns [1998] (see Reyzin [2020] for a survey). Note two differences with the standard SQ model: we look at multiplicative instead of additive errors, and the answers are meaningful only when they are at least the parameter $k$.

We state the algorithms in Section 2.2 assuming robust oracle access. Our implementation of these oracles succeeds with high-probability (over the random sample), hence we give a high-probability cost bound in Theorem 2.5.

## 2.2 Fractional Load Balancing Using Robust Oracle Access

In this section we show how given only robust oracle queries to an unknown instance $\mathcal{I}$, we can compute a parameter vector $\theta$ which achieves a near-optimal load for $\mathcal{I}$, using only poly-logarithmically many queries to this oracle. This will be used in the next section to get our algorithm for load-balancing in the AOS model.

Algorithm 1 starts with an estimate $\theta^0$ of the optimal weight vector (say, the all-ones vector). In each round $t$ it asks for an estimate of the maximum load given the current weights $\theta^{t-1}$; if the load (or rather, the estimate) for some machine is too high, the algorithm reduces the weight on that machine by a factor of $(1 + \varepsilon)$. Lattanzi et al. [2020] showed that this algorithm converges, but the interesting fact of Ahmadian et al. [2021] is that it converges in few rounds of updates, and we will show how this is useful for our purposes.

---

**Algorithm 1:** Robust Algorithm to Compute Weights

**Input:** number of machines $m$, robust oracle for instance $\mathcal{I}$.
1.1 initialize $\theta_v^0 \leftarrow 1$ **for all** $v \in V$
1.2 $\widehat{Z} \leftarrow$ estimate for optimal load $\mathrm{FOpt}(\mathcal{I})$ from $(k, \varepsilon)$-robust oracle
1.3 **for** $t = 1, 2, \ldots, T$ **do**
1.4      $\widehat{\mathbf{X}}^{t-1} = (\widehat{X}_1^{t-1}, \ldots, \widehat{X}_m^{t-1}) \leftarrow$ estimate for loads $L(\mathcal{I}, \theta^{t-1})$ from $(k, \varepsilon)$-robust oracle.
1.5      **forall** $v \in V$ **do**
1.6          **if** $\widehat{X}_v^{t-1} > (1 + \varepsilon)^4 \cdot \widehat{Z}$ **then** $\theta_v^t \leftarrow \frac{\theta_v^{t-1}}{1+\varepsilon}$ **else** $\theta_v^t \leftarrow \theta_v^{t-1}$
1.7 **return** *weight vector* $\hat{\theta} \leftarrow \theta^T$.

---

**Theorem 2.3** (Load Balancing using Robust Queries). *For any $k \geq 0$, let $\gamma := \max(k, \mathrm{FOpt}(\mathcal{I}))$. Fix $\varepsilon \in (0, 1)$ and define $T := 1 + \frac{\ln(2m/\varepsilon) \ln m}{\ln(1+\varepsilon) \ln(1+\varepsilon/2)} = O\big(\frac{\log m \log(m/\varepsilon)}{\varepsilon^2}\big)$. The parameter vector $\hat{\theta}$ returned by Algorithm 1 ensures that the maximum fractional load is*

$$\|L(\mathcal{I}, \hat{\theta})\|_\infty \leq (1 + \varepsilon)^7 \cdot \gamma.$$

If we had direct access to the instance $\mathcal{I}$, we could use directly use the work of Ahmadian et al. [2021]. Since we access $\mathcal{I}$ only though robust oracles, we need a bit more care: we give the short and arguably simpler proof in Section 2.5.

## 2.3 The AOS Algorithm for Load-Balancing

**Lemma 2.4** (Sampling Lemma). *Given any instance $\mathcal{I}$ and $\delta \in (0, 1)$, let $\mathcal{I}_\delta$ be the sub-instance obtained by picking a random $\delta$ fraction of jobs in $\mathcal{I}$. Then*

*i. For $k = \Omega(\varepsilon^{-1} \log m)$, with probability $1 - 1/poly(m)$,*

$$(1 + \varepsilon)^{-1} \leq \frac{\max(k, \text{FOpt}(\mathcal{I}_\delta))}{\max(k, \delta \, \text{FOpt}(\mathcal{I}))} \leq (1 + \varepsilon).$$

*ii. For any fixed weight vector $\theta$, $L(\mathcal{I}_\delta, \theta) \in \delta(1 \pm \varepsilon) \, L(\mathcal{I}, \theta) \pm O(\varepsilon^{-1} \log m)$ with probability $1 - 1/poly(m)$.*

Lemma 2.4 implies that taking a uniformly random subset of $\delta$-fraction of the input and using that to estimate (a) the optimal load or (b) the fractional load $L(\mathcal{I}, \theta)$ for any fixed weight vector $\theta$ gives us a $(k, \varepsilon)$-robust estimate (after rescaling by $1/\delta$) with probability at least $1 - 1/poly(m)$, as long as $k = \Omega(\log m/\varepsilon^2)$.

Let $T$ be the number of rounds required for Algorithm 1, given by Theorem 2.3. If $\text{FOpt}(\mathcal{I}) = \Omega(\frac{T}{p} \frac{\log m}{\varepsilon^2})$, taking the $p$-sample and breaking it randomly into $T$ parts implies that each sample is a $\delta$-sample for $\delta := p/T$. As long as the expected optimal value on each of these parts is much larger than $k$, our estimates are within a factor $(1 + \varepsilon)$ of the correct values. This means that the fractional load of our assignment will also be close to $\text{FOpt}(\mathcal{I})$, as long as $\text{FOpt}(\mathcal{I}) = \Omega((T/p) \cdot (\log m/\varepsilon^2)) = \Omega(\frac{\log^3 m}{p\varepsilon^4})$.

To summarize, the final algorithm is the following:

---
**Algorithm 2:** LEARN-&-LOAD

---
**Input:** Sample $S$ that is a random $p$-fraction of $\mathcal{I}$

**2.1** Randomly partition $S$ into mini-samples $S_1, S_2, \ldots, S_T$ of size $|S|/T$

**2.2** **run** Algorithm 1 but use $S_t$ to estimate $L(\mathcal{I}, \theta^{t-1})$ in each round $t$.

---

**Theorem 2.5.** *For suitably small $\varepsilon \in (0, 1)$, the LEARN-&-LOAD algorithm achieves on any instance $\mathcal{I}$ a fractional load of $(1 + O(\varepsilon)) \cdot \text{FOpt}(\mathcal{I}) + O(\frac{\log^3(m/\varepsilon)}{p\varepsilon^4})$ with probability $1 - 1/poly(m)$, in the AOS model.*

## 2.4 Load-Balancing: Extensions

The above analysis extends with almost no changes to jobs whose sizes $p_v$ lie in $[0, 1]$. Extending to the related machines setting requires a bit more work. Here, each machine $v$ has a speed $s_v \geq 1$, and the load of a machine is the total volume $\sum_e x_{ev}$ assigned to it, divided by the speed. So the goal is to minimize $\max_v (\sum_e x_{ev}/s_v)$. Again, each job can only be assigned to a subset of machines. Keeping the same notation, the machines form a set $V$ of vertices, and the jobs are hyperedges denoting which machines they can be assigned to. In this case, there exist weights $\theta \in \mathbb{R}^m$ such that the scaled proportional allocation

$$x_{ev} := s_v \cdot \frac{\theta_v}{\sum_{u \in e} \theta_u} \cdot \mathbf{1}_{(v \in e)}$$

gives a near-optimal fractional load. Moreover, for each job $e$ we can drop some of the slower machines from the allowed set to ensure that $\frac{\max_{v \in e} s_v}{\min_{v \in e} s_v} \leq (m/\varepsilon)$, at the expense of increasing the optimal fractional load by at most a $1 + \varepsilon$ factor. Moreover, if the speeds of all machines in an instance $\mathcal{I}'$ satisfy $\frac{\max_v s_v}{\min_v s_v} \leq M$ (we say such instances have *aspect ratio $M$*), then the proof of Theorem 2.5 extends to show that $O(\frac{\log^2(Mm/\varepsilon)}{\varepsilon^2})$ rounds of updates suffice. (See the supplementary material for details.) However, this is not enough, since the parameter $M$ can be exponential in $m$, even allowing for $1 + \varepsilon$-approximations.

To handle this, we can perform an instance decomposition: we show how to take any instance $\mathcal{I}$ and find a collection of $K = O(1/\varepsilon)$ "nicer" subinstances $\mathcal{I}^1, \ldots, \mathcal{I}^K$, each of which has aspect ratio bounded by $(m/\varepsilon)^{1/\varepsilon}$, such that each instance contains all the machines, and each job belongs all but one of these $K$ instances. Solving each of these instances $\mathcal{I}^k$ to get assignments $x_{ev}^k$, and then setting $x_{ev} = \frac{1}{K-1} \sum_k x_{ev}^k$ gives a fractional assignment whose load is at most $(1 + 1/(K-1)) = (1 + \varepsilon)$

times the maximum load of each of these subinstances, which is itself near-optimal. The number of rounds increases by approximately a factor of $1/\varepsilon^2$.

We sketch this decomposition, which is of independent interest: define the *speed class* of a machine $v$ to be $c_v := \ln s_v$. Since we assumed that speeds are at least 1, the classes of all machines are non-negative. Assume $1/\varepsilon$ is an integer, for simplicity, For each job $e$ define its *window* $W_e := [\min_{v \in e} c_v, \max_{v \in e} c_v]$. Now for each "shift" $k \in \{0, 1, \ldots, 1/\varepsilon - 1\}$, define $F^k := \{(k + i/\varepsilon) \cdot \ln(m/\varepsilon) \mid i \in \mathbb{Z}\}$, and define $\mathcal{I}^k$ to be set of all jobs $e$ whose window $W_e$ *does not* contain a value in $F^k$. Note that $F^k$ partitions the machines into groups whose speeds classes differ by at most $1/\varepsilon \ln(m/\varepsilon)$, and keeps only jobs whose allowed machine falls within each group. (We can now re-normalize the machine speeds for each group of machines separately, so that the slowest speed in each group is 1, and now the maximum speed is at most $(m/\varepsilon)^{1/\varepsilon}$.) Finally, by our preprocessing each job has a window of width at most $\ln(m/\varepsilon)$, so this window intersects only one of the $F^k$ sets, and hence the job belongs to all but one of the subinstances. This completes the extension of our result to the case of machines having different speeds; the guarantees suffer by only a factor of $1/\varepsilon^2$.

### 2.5  Proof of Theorem 2.3

Finally, we give the proof for Theorem 2.3, mostly to emphasize its simplicity. It relies on two lemmas: a *quasi-monotonicity property* (Lemma 2.6) and an *expansion lemma* (Lemma 2.7). We state these two lemmas here and prove Theorem 2.3, and defer their proofs to the appendix.

Recalling Algorithm 1, let $D_v^t$ denote the number of times that machine $v$'s weight has been decreased by the end of iteration $t$, so that $\theta_v^t = (1 + \varepsilon)^{-D_v^t}$. Define $X_v^t := L(\mathcal{I}, \theta^t)_v$ to be the *actual* load on machine $v$ when using weights $\theta^t$, and let $\widehat{X}_v^t$ be the robust estimate we get in our algorithm.

We first show that if $v$'s weight is decreased at least once then the load is not too small; and if it is not decreased at least once then the load is not too large.

**Lemma 2.6** (Quasi-Monotonicity). *Let $\gamma = \max(k, \mathrm{FOpt}(\mathcal{I}))$. The following guarantees hold:*

> (A) *If $D_v^t > 0$, then $X_v^t \geq (1 + \varepsilon)\gamma$, and*
> (B) *if $D_v^t < t$, then $X_v^t \leq (1 + \varepsilon)^7 \gamma$.*

**Lemma 2.7** (Expansion Lemma). *Let $\alpha = \frac{\ln(2m/\varepsilon)}{\ln(1+\varepsilon)}$. Define $A := \{v \in V \mid D_v^t \geq s\}$ for some $s > 0$, and let $B := \{v \in V \mid D_v^t \geq s - \alpha\}$. Then*

$$|B| \geq \left(1 + \frac{\varepsilon}{2}\right) \cdot |A|.$$

*Proof of Theorem 2.3.* Consider the end of round $T$; we claim that each vertex has $D_v^T < T$. If so, the second claim of Lemma 2.6 guarantees each vertex has load $X_v^t$ at most $(1 + \varepsilon)^7 \gamma$. So, for a contradiction, suppose there do exist vertices with $D_v^T = T$, i.e., whose $\theta$ parameter has been reduced in each round. If we define $M(s) := |\{v \mid D_v^T \geq s\}|$, this means $M(T) \geq 1$. Now applying Lemma 2.7 repeatedly,

$$M(T - j\alpha) \geq \left(1 + \varepsilon/2\right)^j \cdot M(T) \geq \left(1 + \varepsilon/2\right)^j.$$

Hence, if $T > \alpha \cdot \frac{\log m}{\log(1 + \alpha/2)}$, we have $M(1) > m$, which gives a contradiction. This means there are no vertices with $D_v^T = T$, which completes the proof. $\qquad\square$

## 3  Clustering and Network Design in the AOS Model

The online Steiner tree and facility location problems are both fundamental questions in discrete optimization, for which can use the sampling in the AOS model to give an improved performance of $O(\log 1/p)$, whereas the worst-case performance must depend on the size of the instance. For the rest of this section, we assume that $p \leq 1/2$, so that the sample is a small part of the entire instance.

Both these problems are subadditive covering problems: any superset of a solution is another solution; moreover, given any two instances $\mathcal{I}_1, \mathcal{I}_2$, the union of their solutions is also a solution to the union of their instances. This implies that $\mathrm{Opt}(\mathcal{I}_1 \cup \mathcal{I}_2) \leq \mathrm{Opt}(\mathcal{I}_1) + \mathrm{Opt}(\mathcal{I}_2)$. Now for instance $\mathcal{I}$, if $S$ is a $p$-sample for $p \leq 1/2$, and $R$ the remaining requests, this fact implies that $\mathbb{E}[\mathrm{Opt}(R)] \geq 1/2 \cdot \mathrm{Opt}(\mathcal{I})$.

## 3.1 Online Steiner Trees

Let $(V, d)$ be a finite metric space with a fixed root node $r$, where the points $v_1, v_2, \ldots, v_n$ of $V$ are revealed one-by-one. At time $t$ we find out the distances from $v_t$ to the preceding points $v_s$ (for $s < t$), and must connect $v_t$ to one of them. (Think of $v_0 = r$.) The goal is to build a minimum-length spanning tree for $\{v_0, v_1, \ldots, v_n\}$. This problem is often called the *online Steiner tree* problem; the optimal competitive ratio for this problem is $\Theta(\log n)$ due to Imase and Waxman [1991]. In the AOS model, the set $V$ is randomly partitioned into the sample $S$ and the "real" points $R$, where $|S| = pn$. We then see the sample up-front, followed by the points in $R$ in some adversarial order. The goal is to build a minimum-length tree that contains all the points in $R$; it can also contain any subset of $S$. The proposed algorithm is simply an *augmented greedy* algorithm:

> Build a minimum-length spanning tree $T$ connecting the sample $S$. Now if the points in $R$ are denoted $r_1, r_2, \cdots$, then connect $r_t$ to the closest point in $S \cup \{r_1, \ldots, r_{t-1}\}$.

In other words, after we buy an MST on the sample, we simply run the greedy algorithm. (This algorithm is directly inspired by that of Garg et al. [2008] for the case of i.i.d. samples from a known distribution.)

**Lemma 3.1.** *For $p \leq 1/2$, we have $\mathbb{E}[\mathrm{Alg}(R)] \leq O(\log 1/p) \cdot \mathrm{Opt}(V) \leq O(\log 1/p) \cdot \mathbb{E}[\mathrm{Opt}(R)]$.*

*Proof.* The cost of the MST on the sample is $\mathrm{Opt}(S) \leq \mathrm{Opt}(V)$, so it suffices to bound the augmentation cost for requests in $R$. Consider the minimum-length spanning tree $T^*$ for $V$; taking an Eulerian tour and shortcutting repeated vertices gives a cycle $C^*$ of length at most $2\,\mathrm{Opt}(V)$ connecting all points in $V$. The sample $S$ breaks this cycle into paths; for each edge $e \in C^*$, let $L_e$ be the number of vertices on this path containing $e$, and let $d_e$ denote its length.

**Claim 3.2.**
$$\mathbb{E}[\mathrm{Alg}(R)] \leq O\left(\mathbb{E}\left[\sum_e d_e \log L_e\right]\right) \leq O\left(\sum_e d_e \log \mathbb{E}[L_e]\right).$$

*Proof.* The proof is very similar to showing that the greedy algorithm is $O(\log k)$-competitive on sequences of length $k$; we give it here for completeness. Let $P$ be one path in the partition of $C^*$, where $P = \langle s = u_0, u_1, u_2, \ldots, u_{a-1}, u_a = s' \rangle$ with $s = u_0$ and $u_a = s'$ belonging to $S$, and all other $u_i \notin S$. Let $A = \{u_0, \ldots, u_a\}$. Taking alternate edges partitions $P$ into two matchings, each of length at most $\sum_{e \in P} d_e$. Consider the matching $M$ with at least half the edges in $P$: for each edge $e = (u_i, u_j) \in M$, the cost of connecting the later-arriving of the two (say $u_j$) greedily is at most $d(u_i, u_j) \leq d_e$. Hence the total cost for connecting all these later-arriving vertices is at most $\sum_{e \in P} d_e$. This accounts for half the vertices on $P$; now dropping all these later-arriving vertices from $A$, and repeating the argument $\log_2 A$ times shows that the total cost incurred for nodes in $P$ is at most $O(\log_2 |P|) \cdot \sum_{e \in P} d_e$. Finally, summing over all the paths completes the proof of the first inequality. Then using Jensen's inequality along with the concavity of the logarithm function gives the second inequality of the claim. $\square$

Finally, $S$ is a random subset of $pn$ points from $V$ (giving a hypergeometric distribution), so the number of hops from the edge $e$ to the closest vertex in each direction in $S$ is bounded above by a geometric random variable with parameter $p$, which has expectation $1/p$. This means $\mathbb{E}[L_e] \leq 2/p$, and now Claim 3.2 and the fact that $\mathbb{E}[\mathrm{Opt}(R)] \geq \Omega(\mathrm{Opt}(V))$ for $p \leq 1/2$ completes the proof. $\square$

## 3.2 Facility Location

In the facility location problem, again a metric space $(V, d)$ is revealed online; we maintain a set $F \subseteq V$ of *opened facilities* at each time, and for each revealed vertex $v_j$, we can (a) add $v_j$ to the set of opened facilities at cost $f$, or (b) assign $v_j$ to a previously opened facility $i \in F$ at cost $d(v_j, i)$. The objective is to minimize the sum of the facility-opening costs and the assignment costs, namely

$$|F| \cdot f + \sum_{j \in V} d(j, F).$$

Here $d(f, F) := \min\{d(j, i) \mid i \in F\}$.

Our AOS algorithm is the following: given a random sample $S \subseteq V$ with $pn$ clients, let the real set $R := V \setminus S$ be the remaining $(1-p)n$ clients. In the *first stage*, imagine each client in $S$ to have *demand* $1/p$; use a constant-factor approximation to the facility location problem to find $\widehat{F}$ minimizing

$$|\widehat{F}| \cdot f + \sum_{j \in S} d(j, \widehat{F}) \cdot (1/p).$$

Now in the *second stage*, run Meyerson's algorithm on the remaining clients. Specifically, define $F \leftarrow \widehat{F}$; now for each client $j \in R$, let $\delta_j$ be the distance $d(j, F)$. Open a facility at $j$ with probability $\min(1, \delta_j/f)$ (and therefore add $j$ to $F$), else pay at most $\delta_j$ to connect to this facility. Note that the expected cost for each $j \in R$ is at most $\min(1, \delta_j/f) \cdot f + (1 - \min(1, \delta_j/f)) \cdot \delta_j \leq 2 \min(f, \delta_j)$.

**Lemma 3.3.** *For $p \leq 1/2$, we have $\mathbb{E}[\mathrm{Alg}(R)] \leq O(\log 1/p) \cdot \mathrm{Opt}(V) \leq O(\log 1/p) \cdot \mathbb{E}[\mathrm{Opt}(R)]$.*

*Proof.* Consider the cost of the optimal solution $F^*$ for the first stage: the expected cost is at most

$$|F^*| \cdot f + \sum_{j \in V} d(j, F^*) \cdot \Pr[j \in S] \cdot (1/p) = \mathrm{Opt}(V).$$

Hence the constant-factor approximate solution $\widehat{F}$ incurs a cost of at most $O(\mathrm{Opt}(V))$ in the first stage. It now suffices to bound the cost incurred in the second stage, i.e., on the clients in $R$.

Consider the optimal solution for $V$, and let $i^*$ be an open facility which serves some subset ("cluster") of clients; by renumbering, let these clients be $C_{i^*} := \{1, 2, \ldots, m\} \subseteq V$ in order of increasing distance from $i^*$. Let the total connection cost of these clients be $Z_{i^*} := \sum_{j=1}^{m} d(j, i^*)$. Some of these clients $j_1 < j_2 < \ldots < j_k$ belong to the sample $S$; let the facility in $\widehat{F}$ serving client $j_a$ be denoted $i_a \in \widehat{F}$. First, for a client $j \in (j_a, j_{a+1})$,

$$\delta_j = d(j, F) \leq d(j, \widehat{F}) \leq d(j, i^*) + d(i^*, j_a) + d(j_a, i_a) \leq 2\,d(j, i^*) + d(j_a, i_a), \quad (4)$$

since $j_a < j$ and hence its distance to $i^*$ is no larger. By our observation above, the expected cost incurred by such $j$ is at most $2\min(\delta_j, f) \leq 2\delta_j$. Summing over such clients, all indices $a \geq 1$, and over all optimal facilities $i^*$, the first term on the right of (4) sums to at most twice the optimal connection cost; and the summation for the second term has each $j_a$ appear at most $1/p$ times in expectation, so its expected value is at most the connection cost of the first-stage solution, and hence at most $O(\mathrm{Opt}(V))$. Hence the expected connection cost for these clients $j \geq j_1$ is $O(\mathrm{Opt}(V))$; it remains to bound the connection cost of "close" clients $C'_{i^*} := \{1, 2, \ldots, j_1 - 1\}$. We claim that this cost is at most $O(\log j_1) \cdot Z_{i^*}$; now using the fact that $\mathbb{E}[j_1] = O(1/p)$ and Jensen's inequality on the logarithm function, the expected cost is at most $O(\log 1/p) \cdot Z_{i^*}$ for this cluster, which completes the proof.

The proof of the $O(\log j_1) \cdot Z_{i^*}$ bound mimics that given by Meyerson [2001]: let $z$ be the average cost of these "close" clients in the subset $C'_{i^*}$. We group these clients $j$ into annulii based on their $\delta_j$ values being in the intervals $[0, z], (z, 2z], \cdots, (2^{k-1}z, 2^k z], \cdots$; we can restrict the exponent $k$ to at most $\log_2 j_1$, else the connection cost of that single facility would be too high. Let $G$ be the subset of clients in some interval $(2^{k-1}z, 2^k z]$: the expected cost incurred until the first facility is opened (say at location $j'$) in this subset is at most $2f$ after which the expected costs are at most $O(d(j, j') \leq O(d(j, i^*))$. The sum over the clients in $G$ is at most the cost of this cluster in expectation. Hence each annulus costs at most $O(Z_{i^*})$; there are $\log_2 j_1$ annulii, which proves the $O(\log j_1) \cdot Z_{i^*}$ bound. $\qquad\square$

A better competitive ratio of $O(\frac{\log n}{\log \log n})$ using a primal-dual algorithm was given by Fotakis [2008]; we leave it as an open problem to extend our result to give an $O(\frac{\log 1/p}{\log \log 1/p})$ algorithm.

# 4  Conclusions

There are several directions for future work. The current bounds for load-balancing are not tight: our techniques cannot be directly improved, but we hope that future work can improve the bounds on the additive load. While we can extend our results to related machines (where the size of a job $j$ on machine $i$ is $1/s_i$ for some "speed" $s_i$), we do not know how to extend our results (and specifically the expansion lemma) to unrelated machines where each job $j$ has a size $p_{ij} \in [0, 1]$ for machine $i$.

## Acknowledgments

This research was supported in part by NSF awards CCF-1955785, CCF-2006953, CCF-2224718, and DMS-1952285. We thank Marco Molinaro for many useful discussions.

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
