# A  Proofs from Section 2.3

*Proof of Lemma 2.4.* Consider the optimal fractional assignment $\mathbf{X}^*$ for $\mathcal{I}$; for a machine $i$, let the load on this machine be $\lambda$. Now using the same assignment for the random sample $\mathcal{I}_\delta$ gives an expected load of $\mu := \delta\lambda$ on machine $i$, and the probability that this load deviates from the expectation by $\gamma := \max(\varepsilon\mu, k)$ is at most

$$2\exp\left(-\frac{\gamma^2}{2\mu + \gamma}\right).$$

Suppose $\varepsilon\mu > k$ where $k = O(\varepsilon^{-1}\log m)$, this quantity is at most

$$2e^{-\varepsilon^2\mu/3} \le 2e^{-\varepsilon^k/3} \le 1/\mathrm{poly}(m).$$

ELse $k \ge \varepsilon\mu$, and so the probability is at most

$$2e^{-\varepsilon k} \le 1/\mathrm{poly}(m).$$

This proves the lemma. $\qquad\square$

# B  Proofs from Section 2.5

*Proof of Lemma 2.6.* By induction on $t$; for $t = 0$ the value $D_v^t = 0$ and the claims are vacuously true. Hence we consider iteration $t \ge 1$ that generates $\theta_t$ from $\theta_{t-1}$, and look at two cases.

*Case 1:* $D_v^t = D_v^{t-1}$. Since the algorithm did not update the weight for machine $i$ in iteration $t$, we must have had $\widehat{X}_v^{t-1} \le (1+\varepsilon)^4 \cdot \widehat{Z}$. By the estimation guarantee, $\widehat{X}_v^{t-1} \ge (1+\varepsilon)^{-1}X_v^{t-1}$ and $\widehat{Z} \le (1+\varepsilon)\gamma$, so $X_v^{t-1} \le (1+\varepsilon)^6\gamma$. Since all weights are non-increasing and change by at most a $(1+\varepsilon)$ factor, the new load $X_v^t \le (1+\varepsilon)X_v^{t-1}$—at worst, the weight for machine $v$ may remain the same whereas weights for other machines may decrease. Thus $X_v^t \le (1+\varepsilon)^7\gamma$. This proves the second claim.

For the first claim, if $D_v^t > 0$ then $D_v^{t-1} = D_v^t$ means we can use the induction hypothesis to infer $X_v^{t-1} \ge (1+\varepsilon)\gamma$. Moreover, $X_v^t \ge X_v^{t-1}$, since $\theta_v^t = \theta_v^{t-1}$ and all other weights are non-increasing. So we have $X_v^t \ge (1+\varepsilon)\gamma$.

*Case 2:* $D_v^t = D_v^{t-1} + 1$. Since the algorithm updated the weight, $\widehat{X}_v^{t-1} > (1+\varepsilon)^4 \widehat{Z}$. From the estimation guarantee, we have $\widehat{Z} \ge (1+\varepsilon)^{-1}\gamma$, and in particular, $\widehat{Z} \ge (1+\varepsilon)^{-1}k$. This gives $\widehat{X}_v^{t-1} \ge (1+\varepsilon)^3k$. The estimation guarantee now means that $\max(X_v^{t-1}, k) = X_v^{t-1}$, since otherwise we would have $\widehat{X}_v^{t-1} \le (1+\varepsilon)k$. Moreover, the estimation guarantee says $X_v^{t-1} \ge \widehat{X}_v^{t-1}(1+\varepsilon)^{-1}$, so combining the above facts we get $X_v^{t-1} \ge (1+\varepsilon)^2\gamma$. Since the weight $\theta_v^t$ decreases by a factor of at most $(1+\varepsilon)$, while other weights are non-increasing, we have $X_v^t \ge (1+\varepsilon)\gamma$, which proves the first claim.

For the second claim, if $D_v^t < t$, then $D_v^{t-1} < t-1$. By the induction hypothesis, $X_v^{t-1} \le (1+\varepsilon)^7\gamma$. Furthermore, $X_v^t \le X_v^{t-1}$ (since we decreased the weight for machine $v$ by $(1+\varepsilon)$, and at worst the weights of all the other machines can decrease by the same amount, so $X_v^t \le (1+\varepsilon)^7\gamma$ as desired. $\qquad\square$

*Proof of Lemma 2.7.* Since $D_v^t \ge s > 0$ for all $v \in A$, we have $X_v^t \ge (1+\varepsilon)\gamma$ by Lemma 2.6. Thus, it follows that

$$\sum_{v\in A} X_v^t \ge (1+\varepsilon)|A| \cdot \gamma.$$

Let $x_{ev}^t$ denotes the load that job $e$ puts on machine $v$ using weights $\theta^t$; that is,

$$x_{ev}^t = \frac{\theta_v^t}{\sum_{u\in e}\theta_u^t} \cdot \mathbf{1}_{(v\in e)}.$$

This implies that the load $X_v^t = \sum_e x_{ev}^t$. We can now rewrite the LHS as

$$\sum_{v\in A} X_v^t = \sum_{v\in A}\sum_{e\subseteq B} x_{ev}^t + \sum_{v\in A}\sum_{e\not\subseteq B} x_{ev}^t. \tag{5}$$

For a fixed job/edge $e \ni v$ with $e \not\subseteq B$, it follows that there exists an machine $w \in e$ with $D_w^t < s - \alpha$. Since $D_v^t \geq s$, we have

$$x_{ev}^t = \frac{\theta_v^t}{\sum_{u \in e} \theta_t(u)} \leq \frac{\theta_v^t}{\theta_w^t} \leq \frac{(1+\varepsilon)^{-s}}{(1+\varepsilon)^{-(s-\alpha)}} = (1+\varepsilon)^{-\alpha} = \frac{\varepsilon}{2m}.$$

Each of $m$ machines has load at most $\mathrm{FOpt}(\mathcal{I})$, so there are at most $m\,\mathrm{FOpt}(\mathcal{I})$ edges. In particular, $\deg(v) \leq m\,\mathrm{FOpt}(\mathcal{I})$ for all machines $v$, and so it follows that

$$\sum_{v \in A} \sum_{e \not\subseteq B} x_{ev}^t \leq \sum_{v \in A} \frac{\varepsilon}{2} \cdot \mathrm{FOpt}(\mathcal{I}) = \frac{\varepsilon}{2} \cdot |A| \cdot \mathrm{FOpt}(\mathcal{I}). \tag{6}$$

Subtracting (6) from (5),

$$\sum_{v \in A} \sum_{e \subseteq B} x_{ev}^t \geq \left(1 + \frac{\varepsilon}{2}\right) |A| \cdot \mathrm{FOpt}(\mathcal{I}). \tag{7}$$

Finally, we have

$$\sum_{v \in B} \sum_{e \subseteq B} x_{ev}^t = |\{e \in E \mid e \subseteq B\}| \leq |B| \cdot \mathrm{FOpt}(\mathcal{I}),$$

where the second inequality uses that the optimal value is the density of the densest sub-hypergraph. Combining this with (7), we get

$$|B| \cdot \mathrm{FOpt}(\mathcal{I}) \geq \sum_{v \in B} \sum_{e \subseteq B} x_{ev}^t \geq \sum_{v \in A} \sum_{e \subseteq B} x_{ev}^t \geq \left(1 + \frac{\varepsilon}{2}\right) |A| \cdot \mathrm{FOpt}(\mathcal{I}),$$

which yields our desired claim when divided by $\mathrm{FOpt}(\mathcal{I})$. $\qquad\square$

If $d$ is an upper bound on the *degree* of any machine, i.e., the maximum number of jobs that go to any machine, then the same argument shows that it suffices to set $\alpha = \frac{\ln 2d/(\varepsilon\,\mathrm{FOpt}(\mathcal{I}))}{\ln(1+\varepsilon)}$, or the weaker bound of $\alpha = \frac{\ln 2d/\varepsilon}{\ln(1+\varepsilon)}$.

## C   A Concentration Bound

**Theorem C.1** (Concentration Bound). *Let $X_1, X_2, \ldots, X_n$ be independent random variables taking values in $[0,1]$. Let $X := \sum_{i=1}^{n} X_i$, $\mu = \mathbb{E}[X]$ and $U \geq \mu$. For every $\delta > 0$, we have*

$$Pr[X > (1+\delta)U] \leq Pr[X > \mu + \delta U] < \left(\frac{e^\delta}{(1+\delta)^{1+\delta}}\right)^U \leq e^{-(\delta^2 U)/(2+\delta)},$$

*and*

$$Pr[X < \mu - \delta U] < e^{-\delta^2 U/2}.$$

## D   Proofs for Related Machines

In the related machines setting, recall that each machine $v$ has a *speed* $s_v \geq 1$, and the load of a machine is the total volume $\sum_e x_{ev}$ assigned to it, divided by the speed. So the goal is to minimize $\max_v(\sum_e x_{ev}/s_v)$. Again, each job can only be assigned to a subset of machines. Keeping the same notation, the machines form a set $V$ of vertices, and the jobs are hyperedges denoting which machines they can be assigned to.

**Lemma D.1** (Proportional Assignment for Related Machines). *There exist weights $\theta \in \mathbb{R}^m$ such that the scaled proportional allocation*

$$x_{ev} := s_v \cdot \frac{\theta_v}{\sum_{u \in e} \theta_u} \cdot \mathbf{1}_{(v \in e)}$$

*gives a near-optimal fractional load.*

*Proof.* Consider the convex program

$$\max \sum_{ev} (x_{ev} \log(x_{ev}/s_v) - x_{ev})$$
$$\sum_{v \in E} x_{ev} = 1 \qquad\qquad \forall e \in E$$
$$\sum_{e:v \in e} x_{ev} \le L\, s_i \qquad\qquad \forall v \in V$$
$$x_{ev} \ge 0 \qquad\qquad .$$

Now the KKT condition for this implies that

$$\log(x_{ev}/s_v) = -\lambda_v + \mu_e + \nu_{ev}.$$

Now using complementary slackness gives us for each $v \in e$,

$$x_{ev} = s_v \cdot \frac{e^{-\lambda_v}}{\sum_{u \in e} e^{-\lambda_u}}.$$

Setting $\theta_v = \exp(-\lambda_v)$ completes the proof.

Another intuitive way of seeing is to imagine splitting each machine of speed $s_v$ into $s_v \cdot M$ unit-speed copies for some very large $M$. (This factor of $M$ is handle divisibility issues, where $s_v$ values are not integers.) The optimal fractional assignment for this old related machines instance and this new unit-speed instance correspond to each other, up to scaling by a factor of $M$ (and the small additional loss due to divisibility issues, which we put aside for now). Given an optimal weight vector for this unit-speed setting, all the copies of the same original machine can be assumed to have the same weight (by symmetry), and hence the total amount of job $e$ going on copies of machine $v$ becomes the expression above. □

**Bounding Width.** Given any related machines instance $\mathcal{I}$, for each job $e$ define a new job

$$e' := \{v' \in e \mid s_{v'} \ge (\varepsilon/m) \cdot \max_{v \in e} s_v\}.$$

Let $\mathcal{I}'$ be the instance with just these new jobs; by definition $\frac{\max_{v \in e'} s_v}{\min_{v \in e'} s_v} \le (m/\varepsilon)$ for all $e' \in \mathcal{I}'$.

**Lemma D.2.** $\mathrm{FOpt}(\mathcal{I}) \le \mathrm{FOpt}(\mathcal{I}') \le (1+\varepsilon)\,\mathrm{FOpt}(\mathcal{I})$.

*Proof.* Since we constrain each job to go on a subset of its original set of machines, the optimal load can only increase. But by how much? Fix any fractional assignment $\mathbf{X}$ for $\mathcal{I}$. Consider any machine $v$ and consider any job $e$ for which this is the fastest machine in $e$. (Break ties arbitrarily.) Let $e'$ be the new version of $e$ as above: let $\delta_e = \sum_{u \in e \setminus e'} x_{eu}$ be the volume of $e$ going to machines that are not allowed any more in $e'$: move all this volume to $v$. I.e., set $x'_{e'v} = x_{ev} + \delta_e$ for this fastest machine, $x'_{e'u} = x_{eu}$ for all $u \in e'$, $u \ne v$. Now the total load for $v$ increases by at most

$$(1/s_v) \cdot \sum_{e:v=\arg\max_{u \in e}\{s_u\}} \delta_e.$$

This sum is at most the total volume of jobs assigned to machines that are slower than $v$ by a factor $m/\varepsilon$. There are $m$ such machines, and each has load at most $\mathrm{FOpt}(\mathcal{I})$, so the total increase in the load for $v$ is at most $(\varepsilon/m) \cdot m \cdot \mathrm{FOpt}(\mathcal{I})$, as claimed. □