# OpenReview forum: "Learning from a Sample in Online Algorithms"
_NeurIPS.cc/2022/Conference — NeurIPS 2022 Accept_

### Official Review · Reviewer_sgBf · 2022-06-27

**Rating:** 7
**Confidence:** 4
**Soundness:** 4 excellent
**Presentation:** 4 excellent
**Contribution:** 4 excellent

**Summary:**

The paper develops "Online Algorithms with a Sample" for the problems of online makespan minimisation on related machines, online facility location and online Steiner tree.

Online Algorithms with a Sample is a relatively new model that assumes that the an arbitrary instance I is chosen by the adversary and the Algorithm receives a random p-fraction of the instance before having to process the instance in an online fashion. The main idea behind the model is that the p-fraction naturally serves as a training set for the algorithm. The paper provides algorithms within this model that essentially have a performance guarantee independent of the size of the input (which is possible in the classic worst-case online model). The exact bounds can be found on lines 49 and 56 of the paper. The (not so surprising) caveat is that the performance now depends on p -- the size of the fraction of the input that the algorithm has access to. Depending on the particular problem and input this can be quite powerful (for example in the unrelated machines scheduling problem if the optimal makespan is much higher than the number of available machines).

**Questions:**

Am I right about the implicit assumption regarding the upper bound on the job sizes, or am I missing something?

**Limitations:**

Yes, they have.

**Strengths And Weaknesses:**

Strengths:

I really enjoyed reading the paper, it considers a relatively new but very natural model and proves very interesting results. The highlight, is probably the result on the scheduling problem, which combines known observations with a very simple multiplicative weights update algorithm to derive the obtained result. The latter two results are also quite elegant, although perhaps a tad too simple for justifying acceptance on their own.

Weaknesses:

I could only identify minor weaknesses in the paper. First, it seems that there is an implicit assumption on a known upper bound on the job size with respect to the scheduling problem. This should be discussed further. It would be nice to also have a discussion about where the difficulty lies in extending/adapting the results for the unrelated machines setting. Finally, I think the model lends itself very well to experiments and I would be interested to see to what extent the suggested model outperforms classical algorithms.

---

> ### Author Response · Authors · 2022-08-02
> **Rebuttal comments**
>
> Thank you for the encouraging comments.
>
> You are correct about needing an upper bound on the job sizes. The paper assumes that the job sizes are in [0,1], we state this in Section 2.4, but will make this more prominent earlier in the paper (when we first talk about job sizes). The case of more general job sizes seems trickier. Getting a good *fractional* allocation is immediate using our approach (since each job can be split arbitrarily). The problem arises in the online rounding step (Lemma 2.1). We cannot do independent rounding according to the marginals (since the variables would not be [0,1]-bounded, and the concentration bounds get weaker). Generalizing to $B$-bounded jobs would give an additional factor of $B$ in the additive term. But a more sophisticated rounding may work to avoid all dependence on $B$. This is an intriguing question, and while we don't see a solution yet, we will work on it!
>
> We will also add a discussion to the paper about the difficulty in extending/adapting the results for the unrelated machines setting. The main difficulty for unrelated machines comes from the fact that we do not know good "low-round-complexity" algorithms in this setting. Specifically, the Expansion Lemma does not go through, because the size of each job on each machine can be different. We feel that some new ideas may be needed for the unrelated machines case.

---

> > ### Comment · Reviewer_sgBf · 2022-08-08
> > **Thanks**
> >
> > Thanks a lot for the helpful response.

---

### Official Review · Reviewer_qRGt · 2022-07-11

**Rating:** 7
**Confidence:** 3
**Soundness:** 3 good
**Presentation:** 3 good
**Contribution:** 3 good

**Summary:**

The paper looks at the problem of learning in an online setting where instead of seeing all points one by one, they give a sample of the points and then use it as an initial set to achieve better bounds for the problems, the restricted assignment load-balancing problem, the Steiner tree network design problem, and facility location clustering. Assuming a p fraction of the points is observed as a sample, they bound the competitive ratio using a function of p. They provide an algorithm, Learn-&-Load for load balancing which gives a $1+\epsilon$ ratio (plus an additive factor that depends on the logarithm of the number of machines). They also provide algorithms for Steiner tree network design problem and facility location clustering that has a ratio of $O(\log 1/p)$.

**Questions:**

How does this compare to the models where you have some hints or predictions on how the data points would be rather than sampling upfront? Just curious.

**Limitations:**

The paper states the limitations of the methods they have proposed and the non-optimality of the bounds they have provided for the load-balancing problem. They note that the current method they have suggested cannot be directly improved and leave the discovery of the improved bounds as an open problem.

**Strengths And Weaknesses:**

- Originality: The idea of using sampling to pre-learn information and improve competitive ratios is an idea they are borrowing from a earlier work as they have stated. Similar ideas have also been used in some online clustering algorithms as well. The work of the paper is to mainly introduce the algorithms that perform well.  They basically use the idea to solve 3 online learning problems.

- Quality and Clarity: The paper is well written and clearly formatted. It is easy to follow the flow.

- Significance:  This extends the use of the sampling model to cover three interesting problems. However, it is not entirely clear how significant it is to find improved algorithms for this specific set of problems.

---

> ### Author Response · Authors · 2022-08-02
> **Rebuttal comments**
>
> Thank you for the encouraging comments.
>
> And thank you also for the question. There are no prior results known for load balancing with predictions on the data points, as far as we know. For facility location and Steiner tree, there are works with predictions on data. The closest is that of Garg et al. [Stochastic Analyses for Online Combinatorial Optimization Problems, SODA 2008] where they assume that the input consists of i.i.d. draws (and the algorithm has access to this distribution). Their algorithm for Steiner tree is very similar (and our algorithm is inspired by theirs). It builds a solution on a set of samples and then greedily augments with the arriving data points. The paper also considers facility location, again for the setting where data are i.i.d. samples from a distribution, with a different algorithm. Given the difference between the i.i.d. setting of the old paper and online-with-a-sample setting in our paper, the proofs of correctness in both cases are different. (We apologize for the inadvertent omission of this citation, and will definitely add this comparison to the next version!)
>
> A more recent connection --- one that we found after the submission --- is that of Azar et al. [Online Graph Algorithms with Predictions, SODA 2022] where a predicted data set is given, followed by the real data set arriving online. The paper defines a notion of metric error with outliers (essentially the Wasserstein distance with some holdouts) and gives algorithms that behave well with respect to this metric error. Their approach is to use both offline and online algorithms, and the resulting algorithms are more complicated than ours. We think we can use their approach to give an $O(\log 1/p)$ guarantee for Steiner tree as well (by using the sample as their prediction, and considering the expected metric error for a sample), although the algorithm would be more involved than ours. Their approach does not seem to give us the $O(\log 1/p)$ for facility location, because one can construct instances where either the number of outliers, or the metric distance can be large, even with small values of $1/p$. Again, we will add this to the next version.

---

> > ### Comment · Reviewer_qRGt · 2022-08-06
> > **Thanks for the response**
> >
> > Thank you for your response. I will retain my score of 7 as I feel this is a sufficiently good paper for an acceptance.

---

### Official Review · Reviewer_SzP5 · 2022-07-12

**Rating:** 7
**Confidence:** 4
**Soundness:** 4 excellent
**Presentation:** 4 excellent
**Contribution:** 4 excellent

**Summary:**

The paper discusses three problems in the context of online computing with a sample. In this context the online algorithm is presented with a random sample of a $p$-fraction of the input sequence, and then as to serve the rest in adversarial order. The three problems are load balancing (of restricted jobs), facility location, and Steiner tree. In all three cases, the worst case competitive ratio is logarithmic in input sequence length, for fully adversarial sequences. This paper shows substantial improvement for, say, constant $p$. In the case of load balancing, the improvement is mostly if the optimum is not too small. In the other two problems, the improvement is more impressive: it's logarithmic in $1/p$. The methods used in the algorithm for the load balancing case (proportional response and multiplicative weights) are also substantially different from the methods used for the other two problems (augmenting a good solution for the sample, using a standard algorithm for the worst case).

**Questions:**

None.

**Limitations:**

Irrelevant.

**Strengths And Weaknesses:**

This is a very nice paper that gives strong results (mostly for facility location and Steiner tree) in a recently explored model of beyond worst case online computing. The algorithms are simple and natural.

The main weakness is that there are no matching lower bounds. In fact, the authors themselves point out that there is a better worst case result for facility location that has the potential to be used to improve slightly their beyond worst case upper bound.

---

> ### Author Response · Authors · 2022-08-02
> **Rebuttal comments**
>
> Thank you for the encouraging comments.
>
> Indeed, the only tight lower bound of $\Omega(\log 1/p)$ is for the Steiner tree case. There is a small gap for facility location between our $O(\log 1/p)$ upper bound and the $\Omega(\frac{\log 1/p}{\log \log 1/p})$ lower bound for facility location that follows from Fotakis' lower bound for the worst case. But for load balancing, no substantial lower bounds beyond an additive $\log 1/p$ are known for this sampling setting, as far as we know. We hope that our work will spur interest and future work on this problem, thereby closing the gaps.

---

> > ### Comment · Reviewer_SzP5 · 2022-08-08
> > **Ack**
> >
> > I wasn't seriously concerned with the gap between upper bound and lower bound. The paper contributes sufficiently as it stands.

---

### Meta-Review · Area_Chair_VrNU · 2022-08-28

**Recommendation:** Accept
**Confidence:** Certain

**Metareview:**

This paper gives improved bounds for the assignment load-balancing problem, the Steiner tree network design problem, and facility location clustering in the case when a some of the input is given as a sample.  This is a strong paper at the intersection of online algorithms and learning.

**Award:**

No

---

### Decision · Program_Chairs · 2022-09-14

Accept